# Sustainable Public Procurement: From Law to Practice

**Javier Mendoza Jiménez** [1,2,*] **, Montserrat Hernández López** [1] **and Susana Eva Franco Escobar** [3]

1   Department of Applied Economics and Quantitative Methods, Faculty of Economics, Business and Tourism, Universidad de La Laguna, Camino de la Hornera s/n, 38200 San Cristóbal de La Laguna, Spain; mhdezl@ull.edu.es

2   Escuela Universitaria Iriarte, Paseo Santo Tomás, s/n, 38400 Puerto de la Cruz, Santa Cruz de Tenerife, Spain

3   Department of Basic Law Disciplines, Faculty of Law, Universidad de La Laguna, Camino de la Hornera 37, 38200, San Cristóbal de La Laguna, Spain; sefranco@ull.edu.es

*   Correspondence: jmendozj@ull.edu.es; Tel.: +34 699 25 60 65

**Abstract:** This study aims to propose actions to improve the implementation of sustainable public procurement by identifying the problems perceived by public servants and social economy entities. Two types of questionnaires were sent to organizations in Spain and Europe and 217 complete answers were received (152 from the public sector and 65 from the social entities). In addition, 20 semi-structured personal interviews were conducted by phone with managers of social enterprises and four interviews, also by phone, were carried out with relevant people from the public sector. The results of the surveys and the interviews were structured using the analysis of the Strengths, Weaknesses, Opportunities and Threats (SWOT), which was considered consistent with the strategic nature of public procurement. The perceived opportunities for the public sector focus on more efficient use of public resources and improvement of reputation and social equality. For the social entities, more participation in procurement could lead to less dependency on public funds and more visibility. The obstacles for the public sector are related to lack of training and internal resistance to change, as well as, in the case of social entities, to their small size and the tensions with their social object that might derive from bigger competition. The proposed activities focus on two objectives, more training to increase knowledge from the public sector and the readiness of social entities. It is concluded that it is necessary to establish channels of communications between the two groups to avoid possible inefficiencies.

**Keywords:** public procurement; sustainability; social economy; public sector

## 1. Introduction

The role of public procurement to foster sustainable development is gaining relevance both in academic research and in public policies [1]. Public bodies are being encouraged to procure sustainably, to reduce their social and environmental footprint, and to motivate sustainable practices in the private sector [2], however, academic research on the field is not yet sufficiently developed [3].

The inclusion of fundamental social rights in the Treaty of Functioning of the European Union (TFEU) was a decisive step to encourage a more social Europe. The current Title X (art. 151 to 161) of the TFEU allows the sanctioning of legal, reglementary, and administrative dispositions in order to harmonize social policies in the European Union. Public procurement has experienced changes in the last decade toward a more instrumental vision in order to encourage businesses to carry out behaviors which are beneficial for the general interest [4] (pp. 16,17).

The Europe 2020 strategy defined public procurement as a strategy to support sustainable development, overcoming the traditional vision of being a simple expenditure of public funds

to acquire the necessary (at least in theory) goods and services that the public sector needs to carry out its activities.

This "new wave" of public procurement crystalized in the fourth-generation directives, which entered into force in 2014 and emphasize the importance of including social, environmental, and other quality aspects in the different phases of procurement procedures. That does not mean that the price, which has guided most of the public administrations in this field, does not keep its importance as a criterion for decisions.

As with any European directive, the 2014 directives had to be transposed into the different national legislation and the result was a heterogeneous process. Countries such as the UK or Finland followed the basics of European law, while others like Spain or Portugal included a wider framework regarding social and environmental matters [5].

Despite the good intentions of the directives and the national laws for "changing the course" in public procurement, several questions remain unclear about how the legislative changes are going to be implemented in practice. Several organizational factors have been identified as drivers or barriers of sustainable procurement [6] and organizational culture stands out as a particular barrier to sustainable public procurement [7]. Therefore, there is still a lack of knowledge about the problems in the practical implementation of the more recent legislation.

This study, which focused on the case of Spain, aims to identify barriers to the implementation of sustainable public procurement and proposes further courses of action that would facilitate better decision making by the public sector and greater participation of social entities, resulting in the development of more sustainable procurement. To achieve the first aim, it relies on the responses extracted from the surveys and interviews carried out within two target groups, the public sector and the social economy entities. The latter have recently gained importance in public procurement. The Directive 24/2014/EU recognizes that social businesses "might not be able to obtain contracts under normal conditions of competition", [and] "consequently, it is appropriate to provide that Member States should be able to reserve the right to participate in award procedures for public contracts or for certain lots" [8].

In this article, first, we briefly review the changes in sustainable procurement introduced by Directive 2014/24/EU and describe how they have been interpreted by the Court of Justice of the EU. Then, the methodology for the collection of the opinions of the public and the private sector is explained, followed by the analysis of the results obtained. Using the results, a SWOT analysis is performed which serves as the basis for the recommendations and conclusions. Finally, the bibliographic references are provided.

## 2. From Finalist to Strategic, the New Concept of Public Procurement

Public procurement is often seen as the mere execution of a given budget inside a public institution, leaving little room for a proper benchmarking to ensure that the awarded contracts were the best not only in economic terms but also in terms of added value [9]. Some of the barriers to being able to interest citizens in public procurement relate to the constant reorganization of public organizations linked with the electoral cycles [10], which can harm long term processes.

Another factor hindering the reputation of public procurement is the perception of corruption. In the European Union this cannot be associated with the lack of legislation, which includes numerous measures to prevent corruption [11]. To tackle this, the OECD published, in 2009, the document "Principles for integrity in public procurement", which presented 10 basic principles organized in the following four areas: transparency, good government, prevention and monitorization, and responsibility and control [12].

Public procurement is increasingly considered as an instrument to support other public strategies, especially innovation policies [13], which have been conceptualized as innovation public procurement [14]. This new concept of public procurement has been reinforced with the new European Directives approved in 2014. Therefore, public procurement is expected to change its role

from bureaucratic to strategic [15], as well as contribute to social and environmental issues such as promotion of energy efficiency [16], gender equality [17] and fair trade [18]. Two different categories can be distinguished inside this new trend of sustainable public procurement based on the type of considerations to be fostered.

Green public procurement, defined as the introduction of environmental considerations in the processes of public procurement [19], is an important political tool that fosters more sustainable production and consumption models [20] and has been reinforced by various approved legislations [21].

Social public procurement [22], or the socially responsible public procurement [23], connects to the development of the social corporate responsibility of the public sector [24,25]. It has to be accountable for the acquisitions it makes, measuring and controlling the effects they produce. The introduction of social considerations aims, among others, to generate employment opportunities, which is especially important in the context of structural unemployment present in the south of Europe [26]. However, its validity is often questioned due to the necessary link to the subject matter of the contract and the perception that it might harm the procurement general principles [27].

A milestone for sustainable procurement was the case *Gebroeders Beentjes VB vs. the Netherlands*, where the Court of Justice of the EU (CJEU) validated the introduction of a clause for the employment of long-term unemployment persons in the frame of the contract. The sense of this clause was to give an opportunity for employment to the aforementioned collective.

These conclusions were reinforced with the case *Nord-pas-de-Calais vs. the Commission/France* were a similar clause was declared compatible and the principles for the introduction of sustainable criteria were set by the CJEU establishing that the search for the most economic offer "does not preclude all possibility for the contracting authorities to use as a criterion a condition linked to the campaign against unemployment provided that that condition is consistent with all the fundamental principles of community law".

This role of public procurement for sustainable development is acknowledged in the *Guidance on the participation of third country bidders and goods in the EU procurement market* published in 2019 where the Commission links public procurement to the Global Development Objectives of the United Nations and determines that "socially responsible public procurement aims to have a social impact on communities by introducing social considerations in public procurement procedures. It can support sustainable development, contribute to governmental efforts to achieve international sustainability goals" [28].

The bases for sustainable public procurement in Spain are set in Law 9/2017 (LCSP) with several provisions that, in contrast to other national legislations, are not optional but compulsory. Article 1.3 establishes that all contracts must include social, environmental, and innovation considerations, whereas article 202 sets the mandatory introduction of at least one social or environmental condition for the contract. In addition, articles 190 and 191 state the penalties in the case of non-compliance with those conditions.

The LCSP aims for the participation of social entities in public procurement and explores the possibilities of the 2014 directives by establishing a mandatory percentage of reserved procurement for social entities working with people at risk of exclusion and with disabilities. Moreover, it reinforces the consideration of aspects other than price in the procurement of social services, setting the possibility of introducing criteria related to the experience of the personnel and the quality of services among others.

Law 9/2017 includes a nonexhaustive list of social and environmental characteristics that can be used to set either awarding criteria or special conditions in the contract. Article 145, one of the longest in the Law, names, among the social aspects, the promotion of employment and the improvement of labor condition or training in the framework of the contract. On the environmental side, the use of renewable energies, carbon emission reducing mechanisms, and energy and water saving measures are mentioned.

The introduction of all of the above-mentioned criteria are the consequence of the evolution in the legislation in public procurement, which progressively incorporated the social principles of the TFUE in the 2014 directives, and which was later transposed into national laws.

To aid in the development of the legislative mandate of the LCSP, two plans were put in place, in 2019, to determine different measures for implementing the sustainable criteria. The first was the Plan for an Ecological Public Procurement[2] (it can be consulted, in Spanish, at https://www.miteco. gob.es/es/ministerio/planes-estrategias/plan-de-contratacion-publica-ecologica/), which was a literal transposition of the work on Green Public Procurement that the European Commission has been carrying out since 2014. Twenty different products and services were selected mentioning possible characteristics to be valued as awarded criteria or to be introduced as technical specifications. The plan also includes some examples of the redaction of criteria that can be directly introduced in the bid documents by the public buyers (after the necessary adaptation to their specific situation).

On the social side, the Plan to Foster Socially Responsible Public Procurement was published on May 2019[3] (it can be consulted, in Spanish, at https://www.boe.es/diario_boe/txt.php?id=BOE-A-2019-7831). The document is a compilation of the different parts of the law addressing social aspects, but it does not provide guidance for the public administration and it does not provide a single example of possible award criteria or technical consideration. Therefore, its utility in practical terms is, in our opinion, doubtful.

Europe needs sustainable development to reach social, environmental, and economic progress for its citizens, based on cooperation, democracy, and solidarity. In Spain, the LCSP has included some precepts for answering that call. However, as shown in this article, the path to their implementation faces several obstacles, some of them related to the lack of clarity about the validity of certain criteria (especially the social ones) that has not yet been solved by the diverse resolution of the administrative tribunals.

## 3. Methodology

Public employees working on public procurement in Spain were set as the first target group for the study. To ensure the diversity of the answers, all the administrative levels, local, regional, and national, were addressed, as well as diverse regions inside Spain. The surveys (accessible through the link provided in the Appendix A) were sent by email and also filled out in paper, during the realization of diverse training courses and conferences carried out by the authors of this study.

Because one aim of the new legislation is to facilitate the incorporation of Small and Medium Enterprises (SME) and also to favor the participation of social organizations, these were chosen as one of the groups to be consulted on their opinion regarding the possibilities and obstacles to achieve a more sustainable procurement. In addition, taking into account the opinion of the social entities also helps to cover the existing research gap in this field.

To enlarge the geographical scope of the study a partnership was established with the European Network of Social Integration Enterprises (ENSIE) to distribute and coordinate the collection of answers among their members. The ENSIE is present in most of the European countries (not only EU) and encompasses 27 regional and national networks from 23 different countries, which resulted in the questionnaire being translated and sent in Spanish, English, and French (see the Appendix A to access the surveys). The European organizations were analyzed separately from the Spanish ones to assess if the development of the LCSP had influenced the perception of the social enterprises in Spain as compared with those from the European level.

For the case of Spain, two types of social businesses were chosen, the social integration enterprises and the special centers for employment. These two types of entities were considered the most relevant regarding the new processes of public procurement since they are the only ones that can participate in the reserved procurement that the new Law 9/2017 established. Individual emails were sent thanks to the databases provided by umbrella organizations such as the Federation of Association of Social Integration Enterprises (FAEDEI).

Two different surveys with similar aspects were designed, one for each group of potential respondents. The public employees were asked about their training and knowledge in the field; the process for implementing social and environmental aspects in public procurement, including possibilities and obstacles; their relationship and knowledge about the social business; and, finally, their general opinion about the readiness of the public administration for a more sustainable procurement.

Social businesses were given similar questions to collect their opinions about the following: the current legislation, the collaboration with the public sector, the preparation of the social business to be more present on public procurement, and the difficulties and perceived impacts of including social and environmental consideration in procurements.

To add qualitative information to the responses of the surveys, semi-structured personal interviews were carried out with representatives of both groups. The interviews were carried out both by phone and in person. Notes were taken for each of them regarding the region of the entity, the position of the interviewee, and the date, and the duration (although this factor was estimated afterwards since not exact timing was carried out). From the social businesses, 20 people within the managing areas were interviewed. Moreover, four key persons from diverse administrative levels, from the municipality level to the national Congress implicated in the development and implementation of Law 9/2017 were contacted. Table 1 shows the characteristics of the respondents of both groups and the date and type of each interview. The results obtained are explained after the analysis of the surveys of each group.

**Table 1.** Characteristics of the people interviewed.

| Social Entities | | | | |
|---|---|---|---|---|
| Region * | Position of the Interviewee | Type of Interview | Date | Duration (Estimated) |
| Canary Islands | Manager | In person | 20/05/2018 | 10 min |
| Canary Islands | Director | In person | 20/05/2018 | 12 min |
| Canary Islands | Area responsible | In person | 20/05/2018 | 11 min |
| Canary Islands | Director | In person | 20/05/2018 | 10 min |
| Canary Islands | Director | In person | 20/05/2018 | 9 min |
| Canary Islands | Area responsible | In person | 20/05/2018 | 11 min |
| Canary Islands | President of Foundation | In person | 20/05/2018 | 8 min |
| Canary Islands | Manager | In person | 20/05/2018 | 12 min |
| Canary Islands | Director | In person | 20/05/2018 | 10 min |
| Canary Islands | Area responsible | In person | 20/05/2018 | 12 min |
| Catalonia | Director | By phone | 23/01/2018 | 11 min |
| Madrid | Director | By phone | 01/04/2018 | 10 min |
| Madrid | Director | By phone | 01/04/2018 | 9 min |
| Andalusia | Manager | By phone | 17/01/2018 | 7 min |
| Navarra | Area responsible | By phone | 22/02/2018 | 11 min |
| Basque Country | Area responsible | By phone | 01/04/2018 | 13 min |
| Basque Country | Area responsible | By phone | 01/04/2018 | 10 min |
| Madrid | Area responsible | By phone | 23/04//2018 | 14 min |
| Balearic Islands | Manager | By phone | 18/02/2018 | 9 min |
| Castilla-La Mancha | Director | In person | 19/09/2018 | 15 min |
| Public Sector | | | | |
| Name | Type of interview, date and duration (estimated) | Position | Role on public procurement | Expertise |

**Table 1.** *Cont.*

| Social Entities | | | | |
|---|---|---|---|---|
| **Region** [*] | **Position of the Interviewee** | **Type of Interview** | **Date** | **Duration (Estimated)** |
| Javier Tena | Phone 25/01/2018 15 min | Chief of procurement, Madrid Municipality | Implementation | Coordinator of content of the Platform "Contratos del Sector Público" |
| Gustavo Zaragoza | In person 18/04/2018 12 min | Professor, University of Valencia | Research | Responsible for the Plan for a Social Public Procurement in Valencia |
| Luis Bentue | Phone 25/05/2018 13 min | Responsible of the Observatory of Public Procurement, Zaragoza | Consulting | Responsible for the process towards Sustainable Public Procurement in Zaragoza, recognized as a successful example |
| María Jesús Serrano | Phone 04/02/2018 10 min | Congress woman of PSOE, Spanish Congress | Responsible for legislation | Member of the Congress Commission responsible for the redaction and approval of Law 9/2017 |

[*] From the Canary Islands, all the persons were interviewed on the frame of a meeting of WISEs where one of the authors was invited to provide a briefing on reserved contracts. Source: Authors´ elaboration from the notes of the interviews.

## 4. Results

The analysis of the results will be structured as follows: first the opinions of the social enterprises, both at the European and Spanish level, are showed. The second group of answers correspond to the public employees and, finally, a comparison is performed.

### *4.1. Social Enterprises*

#### 4.1.1. European Social Enterprises

A total of 28 complete answers (representing 0,88% of the total number of entities under the umbrella of ENSIE), from which five corresponded to Spanish entities, and therefore have been included in the national group, were received thanks to the partnership with ENSIE. Geographically, nine countries and 15 regions were covered, with Austria being the country with more respondents.

The first set of questions focused on the legal framework and the perceived collaboration with other agents regarding public procurement. As Table 2 shows, the entities are on average satisfied with the influence they had on the approval of the current legislation and clearly pointed out that public servants show insufficient knowledge of legislation in public procurement.

**Table 2.** European social enterprises, satisfaction with legislation and governance framework.

| Group | Variable | Score (On a 7 Points Likert Scale) |
|---|---|---|
| Legislation | National legislation | 3.67 |
| | Regional legislation | 3.20 |
| | Public servants´ knowledge about legislation | 3.05 |
| | Influence of social enterprises in legislation | 3.73 |

**Table 2.** *Cont.*

| Group | Variable | Score (On a 7 Points Likert Scale) |
|---|---|---|
| Collaboration among economic agents | Collaboration between social enterprises | 4.41 |
| | Collaboration with commercial enterprises | 3.64 |
| | Collaboration with public administrations | 4.43 |
| | Professionalization | 5.23 |

Source: Authors´ elaboration from the results of the surveys.

Collaboration with the public administration is valued higher than collaboration with the commercial enterprises (the term "commercial" is used to distinguish the ordinary enterprises from the social ones). This is logical considering that social businesses work with the public sector through other ways of funding such as subventions or special programs that aim to tackle employment and inequality problems, while they compete with regular businesses in the open market.

Moreover, social entities have, in general, a good impression of their professionalization, meaning that from their point of view they are ready for a more relevant role in public procurement.

The second set of questions referred to the potential impact of incorporating social and environmental criteria in public procurement procedures. The entities were asked about their perceptions of the impact of social clauses and reserved markets on their own company, the impact of more sustainable procurement in the social economy of their country, their vision of the compliance with the existing legislation and, finally, their opinion on the potential development of new sectors thanks to the introduction of new considerations.

Figure 1 shows the average for each of the considered factors and indicates that, for the European social businesses, the social clauses have more impact at an individual level than on the whole of the social economy and that this impact is more concentrated on the existing activities of the entities, rather than the development of new sectors.

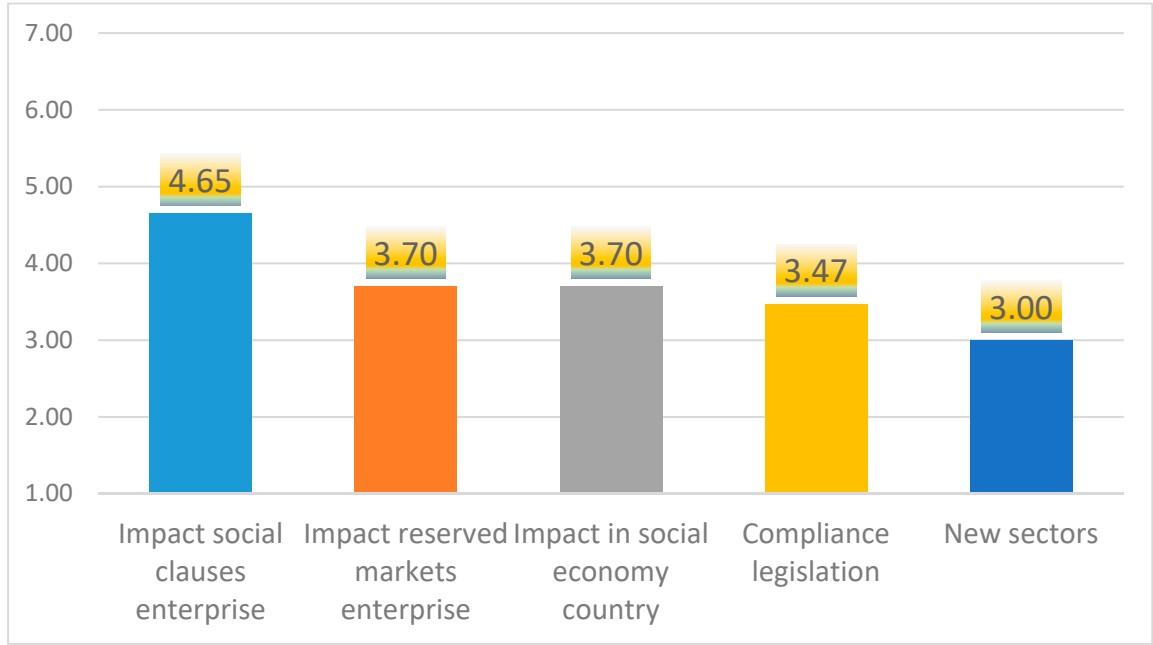

**Figure 1.** European social enterprises, impact of sustainable public procurement. Source: Authors´ elaboration from the results of the surveys.

Meanwhile, the reserved markets are perceived to have a lower potential impact than the social clauses, which can be derived from the fact that, in the majority of national legislation, reserved markets have been kept as an option and not as an obligation, unlike in the case of Spain. It is also remarkable that the low degree of legislation compliance is perceived as an obstacle to a bigger impact from the social point of view.

To add qualitative information regarding the development of sustainable procurement, the final section of the survey included two questions about the perceived opportunities and obstacles related to the introduction of social and environmental clauses. The respondents were asked to give a maximum of two opportunities and two obstacles.

The opportunities for the development of the sustainable procurement can be classified into the following three categories: employment, social entities themselves, and the impact on society in general. With regards to employment, social procurement favors the inclusion of people far from the job market, which is consistent with the existing doctrine in the area.

The social entities perceive an opportunity for more visibility and relevance linked with more access to public procurement. The realization of services, work, and supplies could impact positively on the image of professionalization of these type of entities, which are normally perceived as being connected only with caritative and voluntary activities. From the economic point of view, lower dependence on public funding such as subventions (and the uncertainty associated to them) is another of the positive effects that were pointed out, allowing the social entities to have a more structured plan for their business activities.

Another effect mentioned was the increase of collaboration, both with commercial enterprises which could see the social entities as equal players to collaborate with, and with the public sector. A bigger participation in public procurement would benefit the potential lobby activities of the social economy and would also reinforce its role as a regular provider of services and supplies.

Regarding society in general, and in line with what has been expressed in the European legislation, sustainable public procurement is seen as a vehicle to support poverty reduction and inequalities, and to develop more environmentally sustainable practices.

In terms of the obstacles, public administration, public employees, capitalist enterprises, and the social entities received most of the criticism. Their lack of political willingness for the necessary changes for the model, which translates into insufficient application of the law and corruption, was generally mentioned. In addition, the lack of knowledge by public servants and the complexity of the processes are clear examples of the factors hindering a wider application of the legislation.

It is precisely the legislation that was another focus of complaints, since it is considered to be too complex and subjective, thus provoking legal insecurity. The attitude of commercial enterprises against the development of new ways of procurement was expressed by several of the respondents. Lastly, it is important to note the existence of self-criticism among the social entities, which pointed out the lack of lobbying, the risk of devaluation or their social tasks if they had to compete in pricing, and the lack of legalization and market niches for the collectives at risk of exclusion.

4.1.2. Spanish Social Enterprises

A total of 195 personalized questionnaires were sent to social enterprises to gather their opinion on matters similar to those asked by the European ones, with the addition of a new section that focused on the ease or difficulty of implementing the different social and environmental criteria that the Spanish Law 9/2017 offers. Sixty-five complete answers (i.e., a 33.33% response rate) from 13 of the 17 autonomous regions in Spain were received, with Catalonia and the Canary Islands representing the largest share.

The results obtained in the second section, legal framework and governance of public procurement, showed similarities with those enterprises studied at the European level. As shown in Table 3, the main concern of the social businesses was the low level of knowledge of public servants about the legislation (with a minimum value of 2.42 points on a seven-point scale).

**Table 3.** Spanish social enterprises, satisfaction with legislation and governance framework.

| Group | Variable | Score (on a 7-point Likert scale) |
|---|---|---|
| Legislation | National legislation | 3.35 |
| | Regional legislation | 3.30 |
| | Public servants´ knowledge about legislation | 2.42 |
| | Influence of social enterprises in legislation | 3.62 |
| Collaboration among economic agents | Collaboration social enterprises | 4.38 |
| | Collaboration with commercial enterprises | 2.91 |
| | Collaboration with public Administrations | 3.87 |
| | Professionalization | 3.84 |

Source: Authors´ elaboration from the results of the surveys.

The perceived low level of influence on shaping legislation might be interpreted as a weakness by the social entities and it indicates that they perceive a lack of power as lobbyists. As with the questionnaires at the European level, the most valued collaboration was the relationship between the social economy entities and the public administration, and the relationship with the commercial enterprises was another disturbing point. In addition, the perception of professionalization was lower than at the European level, which could hinder the possibilities of being up to the challenge of having greater participation in public procurement.

With respect to the potential impacts of social considerations, the Spanish enterprises are less optimistic than the European enterprises. As shown in Figure 2, there is a lower level for the impact of social and environmental considerations, both in the enterprises and in the region in general, and, especially, a very poor assessment of the compliance with the recently approved legislation concerning sustainable procurement, which is valued as 2.64 points on a seven-point scale.

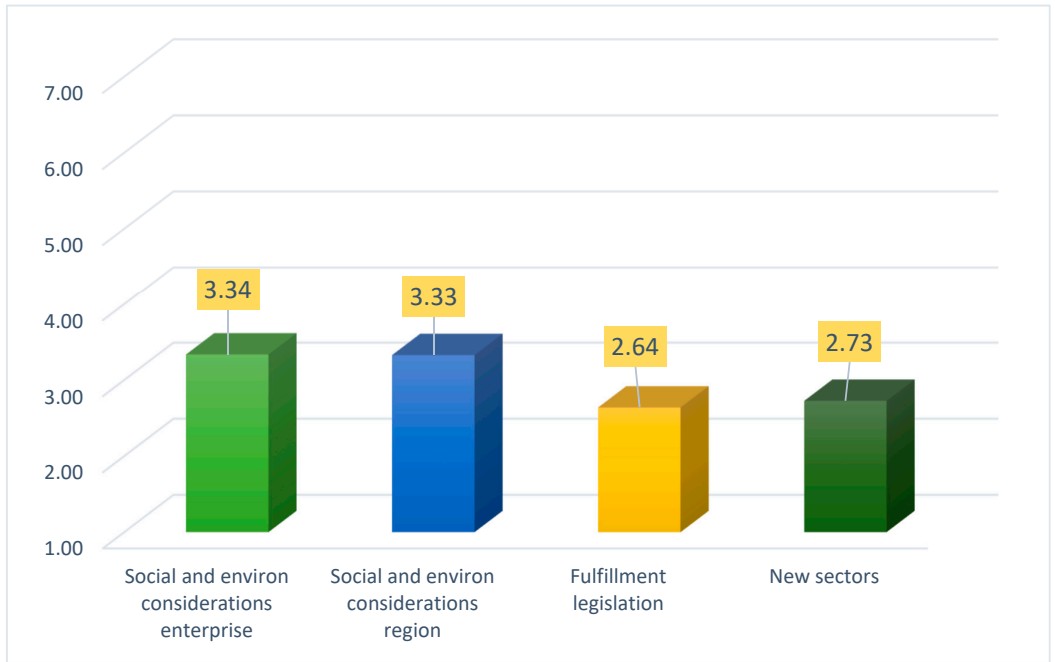

**Figure 2.** Spanish social enterprises, impact of sustainable public procurement. Source: Authors´ elaboration from the results of the surveys.

Therefore, social entities are identified as a greater obstacle than the existence of proper legislation contributing to sustainable procurement. Hence, political willingness could be one of the impediments for the development of such legislation and is further analyzed in depth later in this paper.

Responsible criteria are seen more as a vehicle for consolidation of the existing activities than as a tool to develop new ones. This follows the logic of Law 9/2017, which includes a list of restricted sectors for the reserved procurement based on the traditional activities the social entities carry out in its Annex VI.

A new section of questions was introduced to establish a base of comparison with the opinions of public servants. The aim of the questions was to evaluate the degree of difficulty for introducing a series of social and environmental criteria. To delimit the criteria, the list with the possibilities included in article 145 of Law 9/2017 was used, adding the possibility of reserved procurement.

The results in Table 4 express a certain homogeneity on the perceived difficulty for implementing the criteria, since the majority of them are between 3.5 and 3.9. In general, it can be asserted that, for social entities, the introduction of such considerations should not be difficult. The results for the variation coefficients were below 50% for most of the variables, and therefore rendered the average obtained a representative value.

**Table 4.** Spanish social enterprises, difficulty for the implementation of social and environmental criteria.

| Criteria | Average | Standard Dev | Variation coefficient |
|---|---|---|---|
| Employment | 3.89 | 1.69 | 43.58% |
| Employment at risk | 3.78 | 1.65 | 43.64% |
| Social quality | 3.77 | 1.68 | 44.59% |
| Fair trade | 3.69 | 1.87 | 50.68% |
| Environmental criteria | 3.52 | 1.64 | 46.48% |
| Subcontracting SE | 3.26 | 1.91 | 58.71% |
| Equal opportunities | 3.22 | 1.72 | 53.29% |
| Reserved procurement | 3.00 | 1.44 | 46.15% |
| Average | 3.52 | | |

Source: Authors´ elaboration from the results of the surveys.

In this study, the following three factors were below the average: equal opportunity between women and men, subcontracting with social entities, and reserved procurements. The explanation for the first one might lay in the existence of a consolidated legislation in this field, in Spain, (especially Law 3/2007 for effective equality between women and men) which supports the introduction of these considerations. Moreover, the perception of high professionalization by the social entities reinforces the possibility of establishing subcontracting with them as an award criterion (or even as a condition for the execution of the contract). Additionally, the numerous successful cases for reserved contracts in Spain [5] could encourage their use.

Among the considered criteria, those perceived to be more difficult to implement were related to employment, both in general and in regard to people at risk of exclusion. This could be due to the tendency of using price criterion as the decisive factor in contracts and the controversy that labor issues as social criteria have had in the tribunals. Finally, environmental criteria were considered less difficult to implement than social criteria (on average), reinforcing the idea that there is less litigation involved.

As in the case of the European entities, the questionnaire also included two questions asking about the opportunities and obstacles of a more sustainable public procurement. In comparison to the supranational level, there was a greater variety of answers, both positive and negative, that can be classified into five categories, as shown in Figure 3.

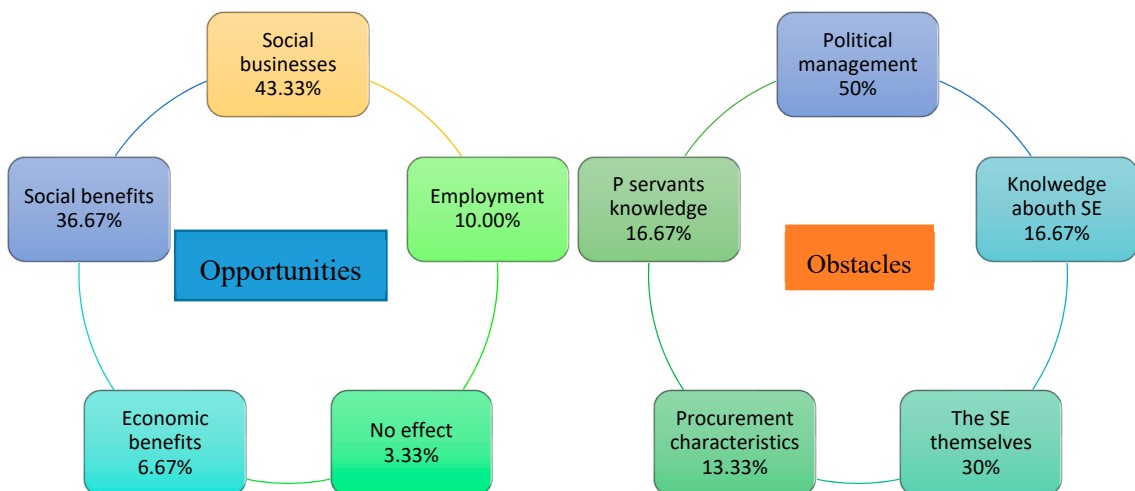

**Figure 3.** Spanish social enterprises, opportunities and obstacles for sustainable public procurement (with % of answers mentioning each one). Source: Authors´ elaboration from the results of the surveys.

Benefits for social businesses were the most frequently mentioned. A more positive image, more visibility, bigger financial security, and presence in the market leading to an increase in competitiveness and employment were highlighted as the potential positive effects. In the social benefits category, more stable employment for people at risk of exclusion and growth in opportunities to join the regulated labor market were highlighted.

Regarding employment, two positive effects were expected, creation of job positions and more professionalization of the people at risk of exclusion and already working. This, in turn, shows an advantage in economic terms both in the increase of the economic return of these type of entities, and in a fairer distribution of public expenditure.

No opportunities linked with commercial entities were mentioned, as was the case for social and environmental consideration leading to more sustainable activities, which indicates a lack of trust in commercial entities from social businesses, and the perception of their attitude as an obstacle for the implementation of the legislation.

In terms of obstacles, public management is the greatest threat to the development of sustainable procurement. Most of the critics focus on lack of political willingness, low interest in change, and a lack of sensibility, knowledge, and awareness. In addition, issues previously mentioned at the European level, such as the incorrect application of the law or the complexity of the law itself, are added to new concerns such as prioritization of price, lack of interdepartmental coordination, and lack of knowledge of the social economy, in general.

Again, self-criticism arose among the social entities, when people pointed at themselves as possible obstacles for social procurement due to their lack of motivation and eagerness to participate more actively in public competition.

### 4.1.3. Social Entities and Personal Interviews

As indicated above, 20 personal interviews were carried out with managers of social entities which were aimed at getting deeper into more qualitative aspects of sustainable public procurement. The aspects addressed through the phone or in person included: preference between social and environmental consideration and reserved procurements, obstacles for sustainable procurement, readiness of social enterprises and public administration, attitude of the commercial businesses, and the forecasted evolution of the sustainable procurement. Managers from six different Spanish regions agreed to personal interviews, with most of them located in Catalonia and the Basque Country.

### 4.1.4. Preference between Social Clauses and Reserved Procurements

There was a similar number of respondents that choose the first and the second option. The argument in favor of reserved procurements identified the greater advantage that they give to social entities, because they secure that only a certain type of organization can participate.

To support social and environmental considerations, the respondents focused on employment, arguing that it is more beneficial for people far from the job market to be contracted directly by commercial enterprises. In addition, social clauses are seen as a way to push professionalization on social entities and achieve the necessary balance between economic survival and social development.

One group of managers pointed out the complementarity of both tools, with both being necessary to achieve "a protocol of sustainable procurement that combines a percentage of reserved procurements with social clauses". Moreover, the two options allow, in the opinion of one of the respondents "a bigger visibility of our job and the people we work with".

### 4.1.5. Obstacles to Sustainable Public Procurement

Although there was a specific question about this topic in the surveys, to obtain a more detailed description this section was also included in the personal interviews. The answers confirmed the aforementioned concerns from the social side, and therefore lack of knowledge among civil servants, as well as lack of political willingness were mentioned in most of the answers.

In the first group, the non-division in lots (that now is compulsory according to article 99 of Law 9/2017) was highlighted, which "may be caused by the lack of knowledge that the administration shows" and diminishes the "possibilities of working together for the social enterprises".

Commercial enterprises also appear as an obstacle, due to the oligopolistic practices and the offers with extremely low prices. According to one of the respondents, the businesses "do not agree with the social considerations, although a positive evolution can be seen, linked with the development of the corporate social responsibility".

### 4.1.6. Readiness of the Public Administration for Sustainable Procurement

The lack of training was pointed out by the majority of the interviewees as one cause for the slow development of sustainable procurement in Spain. In addition, the "prevalence of the economic criteria in the procurement" was another stumbling block, however, positive cases were also mentioned and "some administrations are making a big effort for the change" which is visible through the editing of different guides and recommendations.

### 4.1.7. Readiness of Social Businesses

The degree of preparation of social entities to face more participation in procurement was also discussed. In this regard, there was almost unanimity considering that they have "knowledge on the subject" and are mostly "competitive with the added value of being a social enterprise". Nevertheless, the amount of the contracts can be an obstacle that many entities could not save, due to the fact that they are local and normally small enterprises, which could not reach the threshold for economic and technical solvency.

This raises the question of division in lots as one of the possible points of conflict between public administration and social entities, since it might imply more administrative tasks from the public side and the need for adjustments to coordinate diverse enterprises.

### 4.1.8. Commercial Enterprises

The influence of commercial enterprises over sustainable public procurement does not have a common answer. Half of the interviewees mentioned them as a potential threat, since they oppose any change in procurement and see the sustainable considerations as a breach to free concurrency because they might favor certain businesses.

This does not preclude the existence of channels of collaboration between social and commercial businesses, and some of the respondents mentioned the creation of working groups and protocols for sustainable development. The establishment of temporal unions as enterprises for participation in some contracts shows the possibilities of professionalization of the sector due to more collaboration. In addition, as the sustainable considerations increase their presence, the commercial enterprises are also aware of their importance and seek a way to introduce them in their offerings.

4.1.9. The Evolution of Sustainable Public Procurement

The last question addressed the impact and future of sustainable procurement. Most of the interviewees were optimistic in that sustainable public procurement could contribute by supporting expansion of the working possibilities for people at risk of exclusion and by strengthening the work of social entities and the development of local, nonprofit entities, avoiding speculation and corruption.

Sustainable procurement is expected to continue to grow in the future, with more administration assuming their central role as a social agent and changing their view of procurement from finalist to strategic. Reserved procurements are the preferred tool to do so, since they secure activities currently going on, but the implementation of sustainable procurement still lacks the training of public servants and more stable channels of communication among the different agents.

*4.2. The Opinion of the Public Sector*

4.2.1. Analysis of the Surveys

The same structure as the one used to collect information on the social sector was used to gather the opinion of the public employees. The questionnaire included similar questions to favor comparison and was divided into the following four sections: training and knowledge, implementation of sustainable procurement, obstacles and opportunities, and the relation with the social businesses. A total of 152 complete answers were received from seven different regions in Spain, almost equally distributed between administrative levels. The local level, with 40.30% of the answers, was the most represented.

As pointed out previously, the level of knowledge about the implementation a more sustainable public procurement is a key factor for success. In this context, the opinions expressed reflect quite a pessimistic panorama. As Table 5 shows, the level of knowledge about the legislation among the people who oversee its implementation is relatively low, and so is the knowledge and perceived relationship with the social entities.

**Table 5.** Public sector, knowledge of legislation and social enterprises.

| Group | Variable | Score |
|---|---|---|
| Knowledge of legislation | National legislation | 3.15 |
| | Regional legislation | 3.05 |
| Collaboration of social businesses | Knowledge of social businesses | 3.08 |
| | Relationship public – social sector | 2.98 |

Source: Authors´ elaboration from the results of the surveys.

Therefore, the following two weaknesses are identified: the lack of knowledge about the new legislation, and the lack of knowledge about social businesses as potential partners. The first is directly related to the lack of training received by public servants. Almost 73% of the respondents expressed that they had not received training courses at the time of answering the survey, although practically all of them responded positively to the possibility of attending specialized courses on the issue.

The second part of the questionnaire was grounded on the opinion about the implementation of the sustainable considerations. A question was added about the difficulty of introducing sustainable criteria in each phase of the procurement. The results show similar levels of assessment between the preparation, the awarding, and the execution phase, ranging from 3.83 to 3.93.

The responses about concrete criteria to be introduced (the same posed for the social businesses) show that the public employees perceive more difficulties in introducing social and environmental criteria than the social enterprises (Table 6).

**Table 6.** Public sector, difficulties in the implementation of social and environmental criteria.

| Criteria | Average | Standard Dev | Variation Coefficient |
|:---:|:---:|:---:|:---:|
| Fair trade | 4.38 | 1.56 | 35.61% |
| Reserved procurement | 4.26 | 1.46 | 34.22% |
| Subcontracting | 4.09 | 1.52 | 37.12% |
| Employment at risk | 4.01 | 1.56 | 38.94% |
| Social quality | 3.99 | 1.58 | 39.57% |
| Employment | 3.98 | 1.65 | 41.37% |
| Environmental criteria | 3.66 | 1.74 | 47.57% |
| Equal opportunities | 3.60 | 1.74 | 48.29% |
| Total | 4.00 | | |

Source: Authors´ elaboration from the results of the surveys.

The answers are more homogeneous in this case, with variation coefficients that do not pass the 50% threshold for any of the variables. The more remarkable differences between the two groups reside in the criteria related to fair trade and especially regarding the implementation of reserved procurement, which is valued as almost 1.5 times more difficult by the public side. This can be linked to the lack of knowledge about the entities that could benefit from the reserve and it is one of the factors that can harm their participation in procurement.

Although public servants express that they perceive the implementation of the criteria to be more complex in general, the environmental criteria and those related to equality between women and men continue to be the ones considered to be the easiest to implement.

The third segment of the survey inquired about the administrative levels and the productive sectors where the sustainable practices were more likely to be developed. There is no clear conclusion about the first part, because, as Figure 4 shows, the municipal and provincial levels collect more of the answers in both categories and in many of the cases where the first was considered the easiest level, the second was considered the most difficult, and vice versa.

More unanimity can be found about the sectors in which the sustainable procurement could be more easily implemented. More than half of the respondents considered the tertiary sector as the most ideal contracts. Within the tertiary sector, services that are intensive in work force and have lower requirements for training were mentioned, such as social activities, cleaning, health, gardening, and waste management.

Contrary to this, works and supplies were the activities that respondents expressed as being more difficult to introduce social and environmental criteria. In addition, services that require high specialization such as architecture, consultancy, and those with intense use of technology are also in this group.

The last group of questions entailed the perceived opportunities and obstacles to implementing a more sustainable procurement. Moreover, respondents were asked their opinion about the attitude of commercial enterprises towards social and environmental criteria. In this case, as Figure 5 shows, the responses can be classified in different groups.

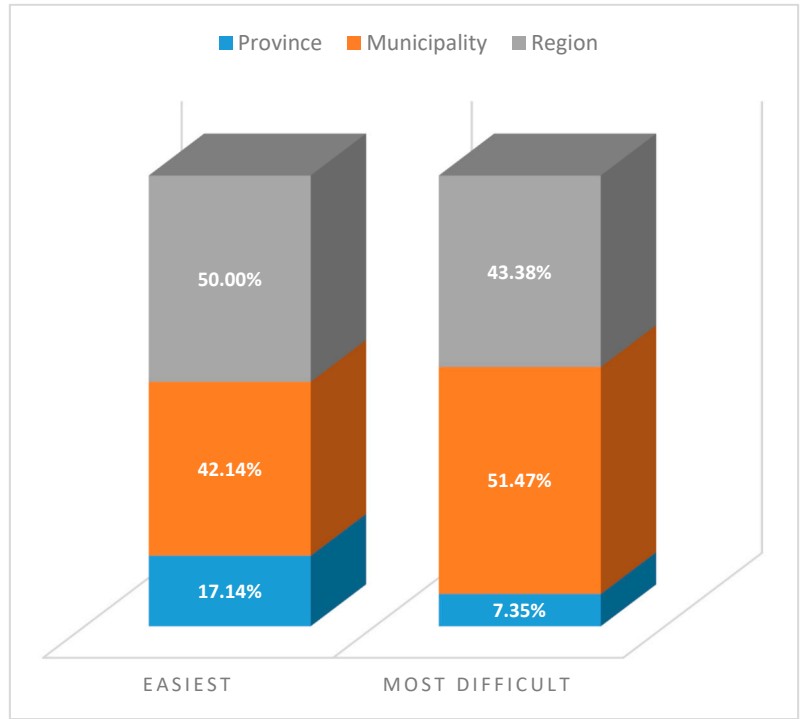

**Figure 4.** Public sector, difficulties in the implementation at the administrative level. Source: Authors´ elaboration from the results of the surveys.

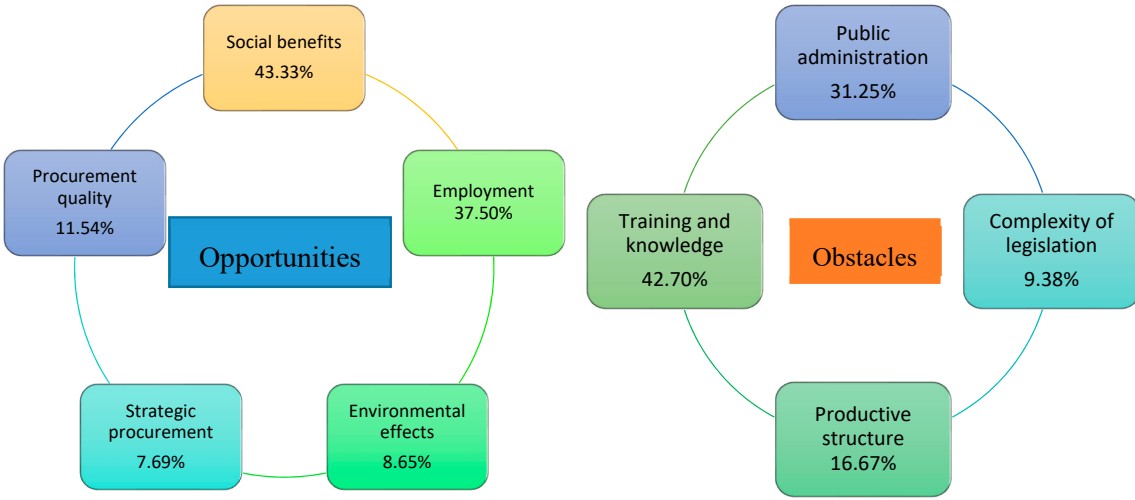

**Figure 5.** Public sector, opportunities and obstacles to sustainable public procurement (with % of answers mentioning each group). Source: Authors´ elaboration from the results of the surveys.

The majority of answers highlighted the potential of sustainable public procurement in areas such as social justice, equality, and better opportunities for people at risk of exclusion. Employment was especially mentioned as well, since sustainable procurement can help to improve labor conditions, wage equality, and labor dignity. Sustainable procurement is seen as a way to increase the quality of public expenditures, balancing social, environmental, and economic aspects and helping change the traditional mentality that links the cheapest with the best. The strategic vision of public procurement can promote a change of culture where social corporate responsibility (private and public) would have more importance, leading to a more positive perception of the citizens about public procurement practices.

On the side of the obstacles, training arises again (this time from the point of view of the public sector) as one of the "stoppers" for the introduction of new criteria. The lack of knowledge relates

to the fear of change, due to the juridical insecurity, and hinders the development of more sustainable practices. Furthermore, as pointed out by the social enterprises, the lack of willingness of the public administrations is frequently mentioned by the respondents, who complain about the immobilism of the public sector and the miss practicing that can lead to corruption and continuation of clientelism.

Nevertheless, factors outside the public sector are also mentioned. The existing productive structure, with a majority of SMEs, a scarce industrial sector, and many businesses which still give low attention to issues beyond the economic ones, are considered barriers to the change. Finally, the legislation itself is said to be complex and unclear, thus, creating insecurity for their application.

### 4.2.2. Personal Interviews

To complete the information obtained from the surveys a group of people from the public sector was interviewed personally or by phone. The selected people came from different administrative levels, i.e., local, regional, and national and were nationally recognized experts on the topic. In contrast to the social enterprises, there was no structured script for the interview. The characteristics of the interviewees and their role in public procurement are presented in Table 1.

The questions posed to the experts in the public sector addressed factors related to their personal experience in the development of more sustainable public procurement. The interviewees were asked to express their opinions about the main obstacles to the implementation of the new social and environmental considerations and the key factors that could hinder or foster it.

Javier Tena considered that public procurement should be more similar to the private one and expressed that the public sector "does not know how to negotiate". Social considerations can help, but there are still many gaps to bridge. For example, the question about the selection and control of the right collective bargain agreement remains unsolved and generates insecurity for public servants.

For Mr. Tena, commercial enterprises "will adapt" to the new obligations if they want to compete in public procurement. In addition, the introduction of social clauses is supportive, but he also recommended performing a previous study on the potential effects they might have and the correct contract to use to introduce them. It is possible that the action would be more efficient if they were implemented, for instance, using the framework of the Social European Fund.

Gustavo Zaragoza highlighted the importance of creating working groups that involve the private and public sectors for the successful implementation of sustainable procurement. In his opinion "social clauses are part of the corporate social responsibility of the public administration in their role to impulse social change with more active social policies".

Municipalities are in a better position to implement sustainable practices, although there is still a long process ahead. Commercial enterprises "do not have normally a positive attitude and some appeals have been made against social criteria, but in general they are starting to accept them".

For Luis Bentue, free concurrency has to be present when developing sustainable criteria. Environmental criteria, for instance, can be formulated in terms of "carbon footprint", favoring proximity alternatives. He recommended establishing a plan for the third sector to improve its visibility and "push the public operators to reserve contracts".

Lastly, Maria Jesús Serrano was satisfied with Law 9/2017, which "goes further than the directives in social and transparency issues". The law could allow social and environmental considerations to have a larger role without eliminating the price criteria. The increased requirements on transparency are not an obstacle to good management.

### 4.3. Comparing Public and Private Sector

The surveys included common questions to expand the comparisons between the two groups into the following three aspects: knowledge on legislation, the difficulty of implementing certain criteria, and the opportunities and obstacles for sustainable procurement.

The low perception of public servants' knowledge about existing legislation is common to both groups and shows the need for more training and information in the public sector. Additionally,

social entities are generally unknown to the public servants, indicating the need for actions from the social sector that raise awareness and establish collaboration channels.

As for the analysis of the implementation of different criteria in the procurement process, Table 7 shows that for all the criteria, the public sector perceives them as more difficulty than the social businesses, with an average of 13.65%.

**Table 7.** Comparison of public and private sector, implementation of criteria.

| Criteria | Value Social Businesses | Value Public Servants | Difference (%) |
|---|---|---|---|
| Employment | 3.89 | 3.98 | 2.31% |
| Employment at risk | 3.78 | 4.01 | 6.08% |
| Subcontracting SE | 3.26 | 4.09 | 25.46% |
| Fair trade | 3.69 | 4.38 | 18.70% |
| Equal opportunities | 3.22 | 3.60 | 11.80% |
| Social quality | 3.77 | 3.99 | 5.84% |
| Environmental criteria | 3.52 | 3.66 | 3.98% |
| Reserved contracts | 3.00 | 4.26 | 42.00% |
| Average | 3.52 | 4.00 | 13.65% |

Source: Authors´ elaboration from the results of the surveys.

The greatest differences can be found for the criteria directly related to social businesses, the subcontracting with them, and the reserved markets. This is especially worrying, taking into account that the implementation of these considerations is linked directly to the lack of knowledge about these organizations by the public sector, who are still not aware of their activities.

Employment, on the contrary, conceals more similar answers, both, in general, and for certain collectives, as does the equality between women and men. The vast legislation in place can be one of the positive supports for this situation. The environmental criteria are perceived to be easier to implement than the social criteria and are in both cases below the average valuation.

Social benefits are mentioned as one of the main opportunities that sustainable procurement could bring. Similarly, employment is also another positive effect when introducing social criteria, and both groups see an opportunity for their own development and reputation that could be derived by putting quality over price.

Both groups agreed that status quo public administration was an obstacle to the advancement of more sustainable practices. Training and knowledge are cited as two of the necessities to be covered and legislation is mentioned, although for two different reasons. From the point of view of public servants, it is complex and lacks clarity, while the social businesses complain about the inadequate application and that the contracts do not adapt to their characteristics.

The productive structure is also highlighted as an obstacle by both groups. For the public administration, there is an insufficient number of enterprises ready for procurement. This is recognized from the social entities when they point out the need for more professionalization and readiness.

Regarding the results derived from the personal interviews, although the structures were different, both groups are in agreement on their optimism about the future development of sustainable procurement, pointing out the necessity for more professionalization in the public sector.

## 5. Discussion: A SWOT Analysis of Sustainable Public Procurement

Some of the key factors for the implementation of sustainable public procurement have been mentioned in this article. For the discussion on the implication of the findings described in the above section, a SWOT analysis was used to organize the findings and give a complete vision.

The SWOT analysis confronts external developments and internal capacities [29] for the later development of new strategies inside the organization [30]. It is a useful methodology to explore

problems from a strategic perspective [31] praised for its simplicity, which has also been considered one of its weaknesses [32]. This methodology has been used in various sectors such as forestry [33], tourism [34], and virtual reality [35].

Looking at the public procurement as a strategic policy, to perform a SWOT analysis with the obtained results could help to understand the external and internal factors conditioning its future development. The analysis, which can be seen in Figure 6, has been built taking into account the two groups of agents inquired and also includes general aspects.

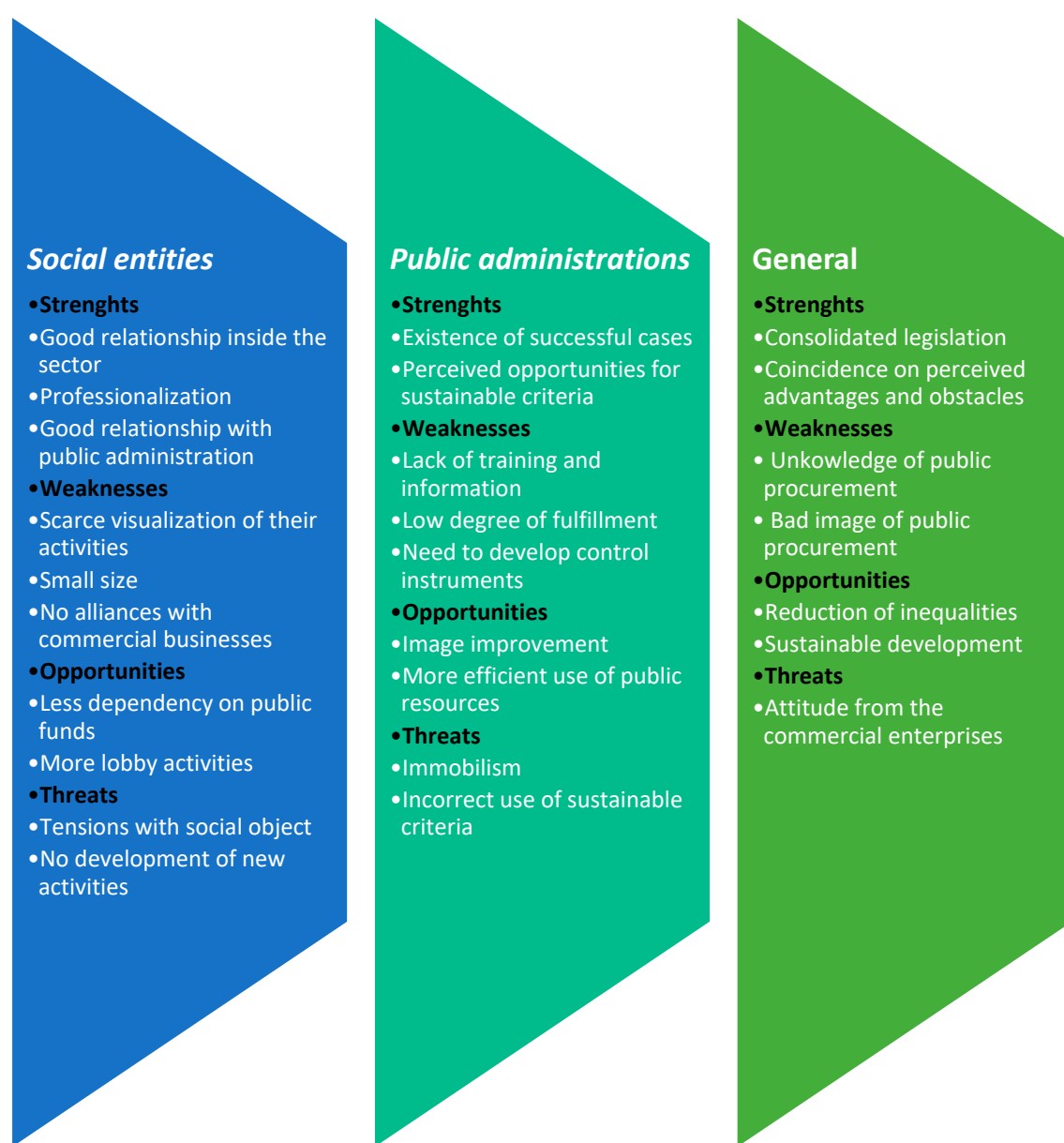

**Figure 6.** SWOT analysis by target group. Source: Authors´ elaboration from the results of the research.

*5.1. Strengths*

Positive relationships, both among social entities and with the public sector, are a strong support for the implementation of sustainable criteria. Some of the more important aspects of Law 9/2017 related to them were introduced thanks to the direct contribution of social groups such as the Confederación Empresarial Española de la Economía Social (CEPES) and the Comité Español de Representantes de

Personas con Discapacidad (CERMI). Professionalization of social businesses is another reinforcement to ensure that they can afford the extra effort needed to participate in public contracts.

The common answers expressed regarding the perceived opportunities and obstacles to implementing public procurement is an advantage to the process. In addition, the public administration has had several successful cases to rely on to advance public procurement. The existing legislation also supports the path of change and, as different legal interpretations shape the law, it will gradually provide more legal security.

*5.2. Weaknesses*

The negative perception about commercial enterprises could hinder the possibilities for establishing networks with social entities, which could benefit from their experience in procurement procedures. The characteristics of social enterprises, mostly SMEs and with low visibility, could prevent the public sector from introducing social and environmental criteria based on the impression that there is no valid interlocutor with whom to discuss them. This is the case for reserved procurements, where the lack of knowledge about the possible participants leads to a greater perceived difficulty in their implementation.

The lack of training and knowledge is probably the greatest weakness that the public administration has to face, and directly affects the development of sustainable procurement. To ensure the right implementation of the new criteria, control and follow-up mechanisms would need to be set up in place.

*5.3. Opportunities*

There are two opportunities for social businesses. First, the opportunity to increase their role as a decisive agent in public procurement, especially in front of the local administration which is one of the biggest buyers. Secondly, the opportunity for more participation in contracts would also lead to less dependency on other public funds such as subventions, which are normally less regular payments and generate more insecurity.

Public administration can find in sustainable procurement an opportunity to improve their image as a generator of wealth and diminish the mistrust in the use of public funds caused by the numerous cases of corruption associated with the awarding of contracts. Creating synergies with social and environmental policies would mean not to expend more but to expend better, increasing the efficiency of the use of public money towards sustainable development.

*5.4. Threats*

According to the public and social sectors, the path for sustainable procurement could find commercial enterprises as an obstacle to overcome. For the social entities, tensions could arise with their social object if they are forced to compete among themselves or in the open market, and economic priorities start to gain more importance. If the opportunities for procurement concentrate in a few sectors, there is also a risk of non-diversification of activities that might endanger them.

For public administration, resistance in order to maintain the current status quo due to lack of political willingness was identified as the main threat. The lack of control and follow-up mechanisms was another important risk for public administration that could result in the misuse of social and environmental criteria or, even worse, put them in bid specifications without controlling that the bidders fulfill their acquired compromises.

## 6. What to Do to Improve the Situation

The findings of this study suggest the type of actions that should be put in place to reinforce public procurement. For the public sector, the lack of knowledge about the legislation and its possibilities makes it necessary to increase not only the level of training in this field, but also the need to establish juridical security on the potential instruments to introduce social and environmental considerations. For the environmental considerations, the work developed by the European Commission on Green Public Procurement, later translated into the Spanish legislation through the "Plan de Contratación

Pública Ecológica" could be used as the base to analyze which environmental aspects of each contract could be implemented.

There is no existing equivalent regarding social considerations, and the recently approved "Plan para una Contratación Pública Socialmente Responsible" is just a collection of the parts of the Spanish Law that address possibilities for inclusion of the social aspects in procurement.

Many of the existing guides and plans lack a key factor to encourage the implementation of these aspects, i.e., they theorize about the possibilities but do not include real examples that would support and ease the work of public servants simply by copying, adapting and pasting. Higher educational institutions should play a greater role in this task, a participation that, so far, seems to be neglected. Their publication of case studies, guides, and similar instruments could make a valuable contribution, descending into the practical field, as well as evaluating the impacts of the introduction of new criteria.

To connect social entities and the public sector, market consultations regulated by article 40 of the 2014/24/EU Directive is another powerful tool that should be put in place. Through those consultations, the public administration could learn the activities that are currently being carried out by the social entities, and therefore determine which are the best activities to enact and adapt the contract requisites to their economic and technical capacity.

The proposed measures for social enterprises focus on increasing their visibility and perception as reliable partners for both the public and the private sectors. The lack of recognition of the work (or even of their existence) manifested by the public sector puts the use of sustainable criteria at risk due to the simple reason of not having someone to collaborate with. Thus, actions destined to increase their work as a social lobbist should be put in place, such as meetings with the public and private sector and partnerships with research institutions to carry out project measuring their contribution to society.

A second line of actions should addresse increasing the professionalization of social entities regarding public procurement processes. The knowledge of the most commonly required documents, the limitations to participate in a contract, and the process of justification of the contract once it is awarded, etc. are capital to avoid mistakes that can result in mistrust from the public administration, and therefore could harm their participation in public procurement. Therefore, a collaborative process should be put in place within the social economy entities, establishing cases of success and, especially, identifying the main difficulties when dealing with a bid process.

## 7. Conclusions

This study offers a vision of the problems and possibilities imbricated on the way towards more sustainable procurement. The results show that there are key factors such as training, the strength of social entities, and the political willingness to build the process. The identification of such elements is fundamental to put in place strategies and instruments for change.

The new laws on public procurement demand an effort from the public administration for greater professionalization, including planning and control, that could generate resistances due to the modification of the current status quo. The support of proven legislative is not enough to automatically introduce the necessary modifications. The traditional association between the lower price and the best bid is one of the main obstacles to change.

To identify the opportunities and obstacles this research introduces a new aspect in academic literature, comparing the opinion of the social entities and the public sector to find ways of collaboration and points of confrontation. The field of analysis was expanded at an international level, thanks to the collaboration with ENSIE, and shows that, at the European and Spanish level, the opinion of social entities is similar, meaning that the LCSP does not have a clear effect on it.

The limitations of the study relate to the low response rate, which made it difficult to get information from both groups. A more in-depth study should be performed, increasing the number of answers, especially in the public sector. Greater regional diversification from the respondents would strengthen the conclusions that have been reached and would allow for the proposal of measures that could be applied state or even Europe wide.

Despite that difficulty, the number of answers can be considered satisfactory, especially for the social aspect where the response rate is normally lower. In addition, the opinions of the public servants, belonging to different levels and regions, enrich the results and allow for comparisons.

This study contributes to the existing literature expanding the field of study that assesses the possible obstacles and possibilities for the implementation of sustainable procurement, in which the majority of the academics have focused on the legislative side of procurement or on the organizational behavior of public administration (see for example [6]). However, to date few researchers have included the opinion of the social economy entities, whose principles are directly linked to the new guidelines for public procurement established in the 2014 Directives and the national legislation. The comparison of both groups creates an opportunity to consider the process towards sustainable public procurement from different points of view and sets the path to build a system of indicators to monitor the process of implementation.

The opportunities for developing more sustainable public procurement are cemented by the existence of legislation supporting it and by cases of success. The positive relationship between public and social entities is also a good starting point. On the contrary, tensions that can arise in the social entities and internal resistances of public servants could harm the process towards a procurement that contributes more to sustainable development and relies less on the price as the saint grail.

The administration of public procurement can no longer be a simple expenditure in a given budget. It must be treated as an investment, and therefore be controlled and examined. Using the words of the Spanish Royal Decree about public procurement announced in 1852, "when celebrating contracts, the public administration cannot aim to get sordid benefits, but to find out which is the real price of the things and pay the fair amount".

**Author Contributions:** Conceptualization, J.M.J., M.H.L., and S.E.F.E.; methodology, J.M.J. and M.H.L.; formal analysis, J.M.J. and M.H.L.; investigation, J.M.J., M.H.L., and S.E.F.E.; resources, J.M.J., M.H.L., and S.E.F.E.; data curation, J.M.J. and M.H.L.; Writing—Original draft preparation, J.M.J., M.H.L., and S.E.F.E.; Writing—Review and editing, J.M.J.

**Funding:** This research was funded by the UPV/EHU, LantegiBatuak, and GEAccounting, grant number US17_24.

**Acknowledgments:** We would like to thank all the people who took the time to answer the surveys and had the patience for the interviews. Our gratitude also to Santiago Rodríguez, Comisionado de Lucha contra la Pobreza y para la Inclusión Social, who supported the distribution of the surveys and believes in a better procurement. Finally, thank you very much to Naomi Álvarez Waló for her corrections and suggestions about the English proficiency.

**Conflicts of Interest:** The authors declare no conflict of interest.

## Appendix A  Appendix

Survey for public servants: available through Google forms here
Survey for Spanish social entities: available through Google forms here
Survey – English - for European social entities: available through Google forms here
Survey – French - for European social entities: available through Google forms here

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
