# Peer review of "Sustainable Public Procurement: From Law to Practice"

_sustainability, doi:10.3390/su11226388_

Round 1
Reviewer 1 Report
The study is interesting and deserves to be published.
Author Response
Dear Sir/Madame,
Thank you very much for your time in reviewing the article and for your kind comments.
Have a nice week!
Reviewer 2 Report
To me the paper Is well structured.
I suggest That two points have to be deepen: please add some details about the regulations on sustainable public procurement in Spain in order to make clear which Is the state of the art in this country as It has bene chosen as a setting of analysis.
Why did you choose other organizations from Europe? Please explain Better why did you choose to keep separate Spain from Europe in your analysis.
Author Response
Dear Sir/Madame,
Thank you very much for your time to review our paper. Regarding your comments we have make the following improvements:
English proficiency: the article has been double checked by a native speaker (her comments can be seen using the tracking function of Word) We have added the most recent regulations in Spain about sustainable public procurement (both the social and environmental part) in order to provide more deep information on the state of the art (page 4 of the manuscript) The European organizations were analyzed separately from the Spanish ones to assess if the development of the LCSP had influenced the perception of the social enterprises in Spain compared to that from the European level. The conclusions of the study remark that there has been no such effect so far since the results of the surveys are similar (page 21).
For any additional information we are at your disposal.
Kind Regards,
Reviewer 3 Report
Dear authors,
many thanks for having the opportunity to review this paper.
In order to improve the paper, I would suggest the following revisions:
improve the structure of the paper. it is not common to have an introduction with a sub-section. I would suggest to have a regular introduction and then a section 2. The paper should highlights the limitations of the study, particularly due to the lack of a statistical random sample for the survey. Besides, some details should be given to the ways in which the interviews were analyzed. In presenting the results, you adopt two different styles in analyzing the interviews of the social entities and the public sector. While for social entities the interviews are analysed according to 5 factors, the same structure is not found in the public sector, which factors did you address in this case? In section 3.3 a comparison is run, based on the survey, but is there any difference emerging from the interviews? if so, please indicate them. In the concluding section, the authors should stress more clearly what is the main contribution of the paper in terms of knowledge advancement, what is added to the extant literature? Most of the comments have policy/managerial implications, which is good, but stress also what the paper adds to the scientific literature. A final proof-editing is required to check for typos, errors.
Author Response
Dear Sir/Madame,
Thank you very much for your very useful comments and time that helped to improve your article.
Following your review we have made the following changes:
English proficiency: the article has been double checked by a native speaker (her comments can be seen using the tracking function of Word) Estructure of the paper: we have introduced the suggested section 2 after the introduction (page 2) Interviews to public servants: The following clarification has been introduced:
- The question posed to the experts in the public sector addressed factors related to their personal experience on the development of more sustainable public procurement. The interviewees were asked to express their opinions on the main obstacles for the implementation of the new social and environmental considerations and the key factors that can hinder or foster it.
- Regarding the results derived from the personal interviews, although the structures were different, both groups coincide on their optimism on the future development of sustainable procurement, pointing out the necessity for bigger professionalization for the public sector.
Limitations: In the conclusion section have been introduce references regarding the limitations coming from the sample used and the necessity to enlarge it to cement the results obtained. Academic contribution: Also in the conclusion section have been remarked the contribution to the existing literature based on the introduction of the point of view of the Social Economy entities and its comparison with the public administration (which is normally the only subject of analysis). It is also highlighted the path initiated to build a necessary system of indicators.
Thank you again for your time, we are at your disposal for any additional clarification you might need.
Kind Regard,